# Thinned Nectarines, an Agro-Food Waste with Antidiabetic Potential: HPLC-HESI-MS/MS Phenolic Characterization and In Vitro Evaluation of Their Beneficial Activities

**DOI:** 10.3390/foods11071010

**Published:** 2022-03-30

**Authors:** Elisabetta Schiano, Vincenzo Piccolo, Ettore Novellino, Maria Maisto, Fortuna Iannuzzo, Vincenzo Summa, Gian Carlo Tenore

**Affiliations:** 1Department of Pharmacy, University of Naples Federico II, Via Domenico Montesano 59, 80131 Naples, Italy; elisabetta.schiano@unina.it (E.S.); vincenzo.piccolo3@unina.it (V.P.); fortuna.iannuzzo@unina.it (F.I.); vincenzo.summa@unina.it (V.S.); giancarlo.tenore@unina.it (G.C.T.); 2Department of Medicine and Surgery, Università Cattolica del Sacro Cuore, 00168 Rome, Italy; ettore.novellino@unicatt.it

**Keywords:** agro-food waste products, abscisic acid, bioactive compounds, glucose homeostasis, polyphenols

## Abstract

Due to the side effects of synthetic drugs, the interest in the beneficial role of natural products in the management of diabetic conditions is growing over time. In the context of agro-food waste products, a screening of different fruit thinning by-products identified thinned nectarines (TN) as the richest matrices of abscisic acid (ABA), a phytohormone with well-documented hypoglycemic potential. These waste-food matrices may represent not only precious sources of ABA but also other bioactive molecules with potential health benefits, such as polyphenols. Therefore, we aimed to perform a qualitative and quantitative characterization of a polyphenolic profile of a TN-based nutraceutical formulation through HPLC-HESI-MS/MS and HPLC-DAD-FLD analyses. Additionally, the in vitro antioxidant and antidiabetic potential of TN was investigated. HPLC analyses allowed us to identify forty-eight polyphenolic compounds, nineteen of which were quantified. Moreover, the results obtained through different in vitro assays showed the antioxidant and antidiabetic potential exerted by the tested nutraceutical formulation. In conclusion, the concomitant presence of different bioactive compounds in TN-based nutraceutical formulation, such as ABA and polyphenols, would reasonably support TN as an innovative nutraceutical formulation useful for the management of glucose homeostasis. Further in-depth animal-based studies and clinical trials are needed to deepen these aspects.

## 1. Introduction

Type 2 diabetes mellitus (T2DM) is a metabolic disease characterized by decreased β-cells insulin secretion and/or insulin resistance, resulting in chronic hyperglycemia (elevated blood glucose levels) [1]. This metabolic disease has reached epidemic proportions: the latest edition of the IDF Diabetes Atlas shows that 10.5% of adults aged 20–79 years are currently affected by diabetes, and this percentage is set to rise even further [1]. Due to the significant individual, social and economic impact of this pathology, the correct management of diabetes represents a primary need, especially in order to avoid microvascular and macrovascular complications related to this condition [2]. The management of T2DM involves a wide range of approaches, such as changes in lifestyle, including diet and physical activity, and pharmacological treatment to achieve metabolic control [3]. Although these approaches can substantially reduce diabetes-related morbidity and mortality, the conventional medications in diabetes treatment can cause unwanted side effects to patients, leading to incompliance and treatment failure [4]. As a result, natural agents from plants and plant products have been the alternative target source for new antioxidant and antidiabetic agents based on their traditional use [5]. Among bioactive molecules of plant origin, a specific interest was recently placed on the investigation of abscisic acid (ABA). In this regard, the literature from the last decades reported the role of ABA as an endogenous hormone produced in humans by pancreatic β-cells, adipocytes and myoblasts in response to glucose [6]. Specifically, the evidence concerning the AMPK-mediated signaling pathways of ABA in glycemic control and the non-overlapping metabolic effects with insulin highlight a leading role of ABA as the first hormonal response in the physiological regulation of plasma glucose levels in humans [7].

Additionally, the role of ABA as a terpenoid phytohormone majorly responsible for the regulation of plant growth and differentiation is widely described by the scientific literature, mainly due to the ability to inhibit germination and promote plant dormancy [8]. A progressive accumulation of this phytohormone is reported during fruit ripening, reaching its maximum concentration at a specific stage after the full bloom and then decreasing to its minimum level at the fruit’s fully ripe/harvest stage [9]. In this scenario, immature fruits derived from the thinning stage would represent rich sources of this phytohormone. Of note, the interest of the nutraceutical industry in agro-food waste products is increasingly growing since they represent still rich sources of bioactive compounds that can be conveniently recovered for the formulation of food supplements useful for the management of diabetic conditions. In the context of fruit by-products, thinned fruits come from the excessive thinning of immature fruits. This process is carried out to improve fruit size and quality in crop management but can lead to a large number of unripe fruits being discarded every year [10]. Regarding their possible industrial applications, unripe fruits are widely used as sources of bioactive compounds to be exploited in food preservation and as functional additives [11]. Nevertheless, thinned fruits have long been underestimated as potential high value-added plant resources, and their physiochemical profile and bioactive capacity remain poorly studied.

Interestingly, a screening of different fruit thinning waste products (i.e., peaches, nectarines, apples, pears and plums) conducted by our research group allowed us to identify nectarines as the richest matrices in terms of ABA content [10]. Thus, a novel nutraceutical formulation based on TN polyphenolic extract was chosen as the ideal candidate to be tested for its hypoglycemic potential. The obtained results showed the ability of this formulation to positively influence postprandial glycemia in healthy human subjects in association with an insulin-sparing mechanism of action [10]. From this point of view, thinned fruits would represent not only precious sources of ABA but also other bioactive molecules with potential health benefits, such as polyphenols. Accordingly, growing scientific evidence describes the potential role of polyphenolic compounds in the treatment of diabetes, as it is able to act on the control of blood sugar on different levels [12,13]. In the light of these considerations, natural products containing bioactive molecules with antioxidant potential would be highly desirable for more efficient management of the diabetic condition. Therefore, in order to investigate high added-value components eventually contained in TN-based nutraceutical formulation, we aimed to perform a qualitative and quantitative characterization of TN polyphenolic profile through HPLC-HESI-MS/MS analysis. Additionally, the in vitro antioxidant and antidiabetic potential of a nutraceutical formulation based on TN was evaluated.

## 2. Materials and Methods

### 2.1. Reagents

All chemicals, reagents, and standards used were either analytical or LC-MS grade reagents. The water was treated in a Milli-Q water purification system (Millipore, Bedford, MA, USA) before use. Gallic acid (purity ≥ 98% HPLC), procyanidin B1 (purity ≥ 90% HPLC), procyanidin B2 (purity ≥ 90% HPLC), procyanidin B3 (purity ≥ 95% HPLC), procyanidin C1 (purity ≥ 90% HPLC), catechin (purity ≥ 98% HPLC), chlorogenic acid (purity ≥ 95% HPLC), neochlorogenic acid (purity ≥ 98% HPLC), caffeic acid (purity ≥ 98% HPLC), vanillic acid (purity ≥ 97% HPLC), syringic acid (purity ≥ 98% HPLC), *p*-coumaric acid (purity ≥ 98% HPLC), epicatechin (purity ≥ 98% HPLC), ferulic acid (purity ≥ 99% HPLC), rutin (purity ≥ 94% HPLC), naringenin (purity ≥ 95% HPLC), quercetin 3-*O*-glucoside (purity ≥ 98% HPLC), kaempferol 3-*O*-glucoside (purity ≥ 90% HPLC), quercetin (purity ≥ 98% HPLC) and reagents for in vitro studies were purchased from Sigma-Aldrich (Milan, Italy).

### 2.2. Sample Collection and Sample Preparation for HPLC Analyses

TN were collected in June 2019 at the orchards of “Giaccio Frutta” society (Vitulazio, Caserta, Italy, 41°10′ N–14°13′ E), at 20–25 days after full bloom, coinciding with the fruit thinning stage. The whole fruits were frozen at −80 °C, freeze-dried and ground to obtain a uniform powder that represented the production batch used for the analysis. For TN polyphenols extraction, 1 g of homogenized sample was suspended in 5 mL of 80% aqueous methanol containing 1% formic acid for 10 min, mixed on a vortex mixer for 1 min, sonicated (Branson Fisher Scientific 150E Sonic Dismembrator) for 10 min, and centrifuged for 10 min at 9000× *g*. The supernatant was decanted, and the pellet was re-extracted with 5 mL of the previous solution. Finally, the combined supernatants were filtered with a 0.22 μm nylon filter (CellTreat, Shirley, MA, USA) and stored at −20 °C until analysis [14].

### 2.3. HPLC Analyses of Samples

#### 2.3.1. Qualitative Polyphenols by HPLC-HESI-MS/MS

An HPLC DIONEX UltiMate 3000 (Thermo Fisher Scientific, San Jose, CA, USA) equipment, coupled with an autosampler, a binary solvent pump and an LTQ XL mass spectrometer (Thermo Fisher Scientific, San Jose, CA, USA), was used for the analysis. The chromatographic analysis was performed according to Maisto et al., with slight modifications [15]. Elution was performed on a Kinetex^®^ C18 column (250 mm × 4.6 mm, 5 μm; Phenomenex, Torrance, CA, USA). The mobile phases were water at 2% formic acid (A) and 0.5% formic acid in acetonitrile and water 50:50 (*v*/*v*). After 2 min hold at 10% solvent B, elution was performed according to the following conditions: from 10% (B) to 55% (B) in 50 min and to 95% (B) in 10 min, followed by 10 min of maintenance; then the column was equilibrated to the initial conditions for the remaining 10 min to recondition the column. The separation conditions were as follows: column temperature was set at 30 °C, inject volume was 10 µL, the flow rate was set at 1 mL/min. The source was a heated electrospray interface (HESI), operated in negative ionization with full scanning (FS) and data-dependent acquisition (DDA). Collision-induced fragmentation was made using argon, with a collision energy of 35.0 eV. The ion source was set using the following parameters: sheath gas flow rate: 30; auxiliary gas flow rate: 10; capillary temperature: 320 °C; source heated temperature: 150 °C; source voltage: 3 kV; source current: 3.20 µA; capillary voltage: −20 V; tube lens: −106.40 V.

#### 2.3.2. Quantitative Polyphenols Analysis by HPLC-DAD-FLD

An HPLC Jasco Extrema LC-4000 system (Jasco Inc., Easton, MD, USA), coupled with an autosampler, a binary solvent pump, a diode-array detector (DAD) and a fluorescence detector (FLD). The chromatographic analysis was performed according to Maisto et al., as previously described [16]. Phenolic acids, hydroxycinnamic acids, flavanols and flavanones were monitored at 280 nm, while flavonols were monitored at 360 nm. Procyanidins were monitored by a fluorescence detector that was performed with an excitation wavelength of 272 nm and an emission wavelength of 312 nm. Peak identifications were based on retention times and standard addition to the samples. Compounds were quantified according to a calibration curve made with six different concentrations over a concentration range of 0.1–1000 ppm and triplicate injections at each level.

#### 2.3.3. HPLC-DAD-FLD Method Validation

Method validation was performed according to the ICH validation guideline (ICH.Q2[R1], 1995), which included the evaluation of a set of parameters, such as linearity range, the limit of detection (LOD), the limit of quantification (LOQ), precision and accuracy [17]. Data relating to method validation are reported in Appendix A. As regards polyphenolic quantification, the construction of a six-point calibration curve using diluted standard solutions was performed. Eight polyphenols (gallic acid, catechin, epicatechin, chlorogenic acid, procyanidins B1 and B2, quercetin and rutin) were selected for method development and validation. In order to assess these parameters, calibration curves were prepared in triplicate. The LOD and LOQ were calculated using the following equations: LOD = 3.3 *S_a_*/*b* and LOQ = 10 *S_a_*/*b*, where *Sa* is the standard deviation of the intercept of the regression line and *b* is the slope of the calibration curve. The precision of the method was evaluated through the percentage coefficient of variation (*CV*%), while method accuracy was evaluated through bias. The intra-day precision and accuracy were assessed with three concentration levels in one day. The inter-day precision and accuracy were assessed with three concentration levels over three consecutive days.

### 2.4. Total Phenol Content (TPC)

The total phenol content (TPC) was determined through Folin–Ciocalteau’s method, using gallic acid as standard (Sigma-Aldrich, St. Louis, MO, USA). Briefly, 0.125 mL of sample (properly diluted with water to obtain an absorbance value within the linear range of the spectrophotometer) underwent an addition of: 0.5 mL of distilled water, 0.125 mL of Folin–Ciocalteau’s (Sigma-Aldrich, St. Louis, MO, USA) reagent and 1.25 mL of an aqueous solution of Na_2_CO_3_ 7.5% (*w*/*v*%), bringing the final volume to 3 mL with water. After mixing, the samples were kept in the dark for 90 min. After the reaction period, the absorbance was measured at 760 nm using a V-730 UV-visible/NIR spectrophotometer operated by Spectra Manager^™^ Suite (Jasco Inc., Easton, MD, USA). Each sample was analyzed in triplicate, and the concentration of total polyphenols was calculated in terms of gallic acid equivalents (GAE) [14].

### 2.5. Antioxidant Activity

#### 2.5.1. DPPH^•^ Radical Scavenging Assay

The antioxidant activity of TN was measured with respect to the radical scavenging ability of the antioxidants present in the sample using the stable radical 2,2-diphenyl-1-picrylhydrazyl (DPPH) (Sigma-Aldrich St. Louis, MO, USA). Briefly, 200 µL of the sample was added to 1000 µL of a methanol solution of DPPH (0.05 mM). The antioxidant effect was evaluated by following the decrease in UV absorption at 517 nm with a UV-visible spectrophotometer (Jasco Inc., Easton, MD, USA). The absorbance of DPPH radical without antioxidant, i.e., the control, was measured as blank. All determinations were in triplicate. Inhibition was calculated according to the formula:[(A_i_ − A_f_)/A_c_] × 100,(1)
where A_i_ is the absorbance of the sample at t = 0, A_f_ is the absorbance after 9 min, and A_c_ is the absorbance of the control at time zero. The 6-hydroxy-2,5,7,8-tetramethylchroman-2-carboxylic acid (Trolox) was used as a standard antioxidant, and the results were expressed in µmol Trolox Equivalent (TE). Furthermore, the results were also reported as EC_50_, which is the amount of antioxidant necessary to decrease the initial DPPH^•^ concentration by 50% [18].

#### 2.5.2. TEAC (Trolox Equivalent Antioxidant Capacity) Assay

The method is based on the ability of antioxidant molecules to quench ABTS^•+^ radical (2,20-azinobis(3-ethylbenzotiazoline-6-sulfonate)), a blue-green chromophore with characteristic absorption at 734 nm. The assay was performed according to the method described by Babbar et al. (2011) [19], with slight modifications. ABTS solution was prepared by dissolving 2.5 mL of ABTS 7.0 mM solution and 44 µL of potassium persulfate 140 mM solution, which was left to react for at least 7 h, at 5 °C in the dark. Then, ethanol–water was added to the solution until an absorbance value of 0.700 (0.05) at 754 nm (Jasco Inc., Easton, MD, USA). The assay was performed by adding 1 mL of diluted ABTS working solution to 100 µL of the sample. The determination of sample absorbance was accomplished after 2.5 min of reaction. All determinations were in triplicate. Blank was performed with ethanol in each assay. Inhibition was calculated according to the formula:[(A_i_ − A_f_)/A_c_] × 100,(2)
where A_i_ is the absorbance of the sample at t = 0, A_f_ is the absorbance after 6 min and A_c_ is the absorbance of the control at time zero. Trolox was used as a standard antioxidant. The results were expressed both as µmol of TE and EC_50_, which is the amount of antioxidant necessary to decrease the initial ABTS^•+^ concentration by 50% [18].

#### 2.5.3. Ferric Reducing/Antioxidant Power (FRAP) Assay

When a Fe^3+^-TPTZ complex is reduced to the Fe^2+^ form by an antioxidant under acidic conditions, an intense blue color with absorption maximum develops at 593 nm [18]. Therefore, the antioxidant effect (reducing ability) of the TN sample was evaluated by monitoring the formation of a Fe^2+^-TPTZ complex with a spectrophotometer (Beckman, Los Angeles, CA, USA). The assay was performed according to Benzie and Strain (1996) and Surveswaran et al. (2007) [20,21], with some modifications. The FRAP assay reagent was prepared by adding 10 vol of 0.3 M acetate buffer, pH 3.6 (3.1 g sodium acetate and 16 mL glacial acetic acid), 1 vol of 10 mM TPTZ prepared in 40 mM HCl and 1 vol of 20 mM FeCl_3_. All solutions were used on the day of preparation. The mixture was pre-warmed at 37 °C. This reagent (2.85 mL) was mixed with 0.15 mL diluted test samples similar to those used for the ABTS and DPPH assays. The mixture was shaken and incubated at 37 °C for 4 min, and the absorbance was read at 593 nm (Jasco Inc., Easton, MD, USA). The blank is represented by the only reagent solution. Blank absorbance must be subtracted from the absorbances of the samples. All determinations were in triplicate. A standard curve was made with Trolox, and the results were expressed as µmol TE.

### 2.6. Antidiabetic Activity

#### 2.6.1. α-Amylase Inhibitory Assay

The α-Amylase inhibitory assay was performed using a modified procedure by Schisano et al. (2019) [22]. A total of 40 µL of TN extract at different concentrations was placed in a plastic tube, adding 160 µL of distilled water and 200 µL of α-amylase solution (4 U/mL). This solution was pre-incubated at 37 °C for 10 min. The starch solution (1% *w*/*v*) used as the substrate was prepared by boiling potato starch in 0.02 M sodium phosphate buffer (pH 6.9). An amount of 400 µL of the above-mentioned starch solution was added to the reaction mixture, then further incubated at 37 °C for an additional 20 min. The reaction was terminated by adding 200 µL of DNS solution (20 mL 96 mM 3,5-dinitrosalicylic acid, 12 g sodium potassium tartrate in 8 mL of 2 M NaOH and 12 mL deionized water). The tubes were then incubated in boiling water for 5. After cooling the tube, the reaction mixture was diluted with distilled water (2 mL). Finally, the sample’s absorbance was read at 540 (Jasco Inc., Easton, MD, USA). A control and a blank were prepared using the same procedure, replacing extract and enzyme, respectively, with distilled water. Acarbose, a well-known antidiabetic medicine, was used as a positive control. Three sets of experiments were conducted for the test: standard, blank and control. The α-amylase inhibitory activity was calculated as follow:Inhibition (%) = {[(A_c_ − A_cb_) − (A_s_ − A_sb_)]/(A_c_ − A_b_)} × 100,(3)
where A_cb_ is the absorbance of the control blank, A_c_ is the absorbance of the control, A_sb_ is the absorbance of the sample blank and A_s_ is the absorbance of the sample. The results were reported as IC_50_, which is the amount of sample necessary to decrease the initial α-amylase activity by 50%. IC_50_ values were determined by plotting percent inhibition (*Y*-axis) versus log_10_ extract concentration (*X*-axis) and calculated by logarithmic regression analysis from the mean inhibitory values [23].

#### 2.6.2. Advanced Glycation End-Product (AGE) Inhibition

The inhibition of AGE formation by TN extract and the standard phenolic rutin was measured through the method described by Justino et al. (2019) [24], with slight modifications. An amount of 500 µL of progressive dilutions of the samples (0.075–70 mg/mL of final concentrations for TN and 0.05–2 mg/mL for rutin) was prepared in distilled water, then added to an assay mixture containing 500 µL of bovine serum albumin (50 mg/L), 250 µL fructose (1.25 mol/L) and 250 µL of glucose (25 mol/L). All the components of the reaction mixture were solubilized in phosphate buffer (200 mmol/L; pH 7.4), containing sodium azide (0.02% *w*/*v*). The solution was incubated at 37 °C for 7 days; thereafter, fluorescence was measured at an excitation wavelength of 355 nm and an emission of 460 nm (Perkin-Elmer LS 55, Waltham, MA, USA). Distilled water was used as a negative control, while the blank was carried, substituting fructose and glucose with phosphate buffer. The inhibitory activity was calculated as a percentage of glycation inhibition (GI) by using the following formula:GI (%) = [(F_s_ − F_sb_)/(F_c_ − F_cb_)] × 100,(4)
where F_s_ is fluorescence intensity in the presence of sample; F_sb_ is fluorescence intensity in the absence of fructose and glucose; F_c_ is fluorescence intensity in the absence of sample; and F_cb_ is fluorescence intensity in the absence of sample, fructose and glucose. Finally, the results were reported as EC_50_.

### 2.7. Statistics

Unless otherwise stated, all the experimental results were expressed as the mean ± standard deviation (SD) of three determinations. Graphics and IC_50_ values determination were performed using GraphPad Prism 8 software.

## 3. Results

### 3.1. Qualitative Polyphenols Analysis by HPLC-HESI-MS/MS

TN polyphenolic extract was characterized by HPLC-HESI-MS/MS. Based on a comparison with the literature data, 48 compounds were putatively identified. A total of 27 phenolic acids, 10 flavans and 11 flavonolds are displayed in Table 1. The identity of 19 compounds (3, 6–9, 13, 14, 16, 20, 24, 28, 29, 32, 34, 36, 37, 38, 43, 45) was verified by comparison with analytical standards.

#### 3.1.1. Phenolic Acid Identification

Phenolic acids represent the most common class of polyphenols. They exhibit a generic fragmentation pattern, with a prominent fragment ion produced by the neutral loss of CO_2_, due to the cleavage of the carboxylic acid group. The neutral losses of H_2_O and CO, typical of the phenolic group cleavage, and the loss of sugar or organic acid for phenolic acid derivatives generate other diagnostic ions. Compound 1 showed a [M-H]^−^ ion at *m*/*z* 195 and a tandem mass spectrum characterized by a base peak ion at *m*/*z* 167 [M-H-CO]^−^ and other fragment ions at *m*/*z* 177 [M-H-H_2_O]^−^, *m*/*z* 151 [M-H-CO_2_]^−^ and *m*/*z* 133 [M-H-CO_2_-H_2_O]^−^. Based on this fragmentation pattern, compound 1 was putatively identified as hydroxycaffeic acid [25]. Compound 2 displayed a [M-H]^−^ ion at *m*/*z* 153 and a base peak ion at *m*/*z* 125, corresponding to the loss of CO. The minor fragment ions at *m*/*z* 135 [M-H-H_2_O]^−^, *m*/*z* 109 [M-H-CO_2_]^−^ and *m*/*z* 97 [M-H-2CO]^−^ suggested the presence of a carboxylic acid group and two phenolic groups. Therefore, compound 2 was tentatively identified as dihydroxybenzoic acid [25]. Compound 3 showed a [M-H]^−^ ion at *m*/*z* 169 and two fragment ions at *m*/*z* 141 [M-H-CO]^−^ and *m*/*z* 125 [M-H-CO_2_]^−^. According to the literature data, this compound was identified as gallic acid [26]. Compound 4 produced a [M-H]^−^ ion at *m*/*z* 343 and a base peak ion at *m*/*z* 297, corresponding to the loss of water and CO. The minor product ions at *m*/*z* 299 [M-H-CO_2_]^−^ and at *m*/*z* 181 [M-H-Hex]^−^ revealed the occurrence of free carboxylic acid and a hexoside group, respectively. Consequently, compound 4 was putatively recognized as homovanillic acid *O*-hexoside [27]. Compound 5 displayed a [M-H]^−^ ion at *m*/*z* 191 and a base peak ion at *m*/*z* 147 [M-H-CO_2_]^−^. The base peak ion and the fragment ions at *m*/*z* 173 [M-H-H_2_O]^−^, *m*/*z* 155 [M-H-2H_2_O]^−^ and *m*/*z* 111 [M-H-CO_2_-2H_2_O]^−^ highlighted the presence of free carboxylic acid and an alcoholic group. The absence of a prominent ion caused by CO loss, typical of the phenolic moiety, allowed distinguishing the alcoholic aliphatic group from the phenolic scaffold. Based on the literature data, compound 5 was tentatively identified as quinic acid [50]. The presence of two caffeoylquinic acids (compounds 6 and 13) was confirmed by the [M-H]^−^ ion at *m*/*z* 353 and two fragment ions at *m*/*z* 191 and *m*/*z* 179, corresponding to quinic acid and caffeic acid ions, respectively. The identification of these compounds was supported by comparison with the authentic standards. In this way, compounds 6 and 13 were identified as neochlorogenic acid and chlorogenic acid, respectively [29]. Compound 10 showed a [M-H]^−^ ion at *m*/*z* 325 and a base peak ion at *m*/*z* 289 [M-H-2H_2_O]^−^ due to the loss of two molecules of water. Other fragments at *m*/*z* 307 [M-H-H_2_O]^−^, *m*/*z* 163 [M-H-Hex]^−^ and *m*/*z* 119 [M-H-Hex-CO_2_]^−^ suggested the presence of a carboxylic acid and a hexoside group. The fragment ion at *m*/*z* 163 matched to coumaric acid ion. According to this fragmentation pattern and the literature data, compound 10 was putatively identified as coumaric acid *O*-hexoside [33]. Four coumaroylquinic acid isomers (compounds 11, 18, 19 and 21) were tentatively detected. Their tandem mass spectra displayed a [M-H]^−^ ion at *m*/*z* 337 and characteristic fragment ions at *m*/*z* 191 [QA-H]^−^, *m*/*z* 173 [QA-H-H_2_O]^−^ and *m*/*z* 163 [M-H-QA]^−^. Based on the literature data, the type of base peak ion allowed to distinguish the connectivity between coumaric acid and quinic acid groups. The 3-*O*-coumaroylquinic acid can be identified by a base peak ion at *m*/*z* 163 [M-H-QA]^−^, while 4-*O*-coumaroylquinic acid produced an intense fragment at *m*/*z* 173 [QA-H-H_2_O]^−^. Therefore, compounds 11 and 18 were putatively identified as 3-*O*-coumaroylquinic acid and 4-*O*-coumaroylquinic acid, respectively [34]. Fragmentation of 5-*O*-coumaroylquinic acid produced a base peak ion at *m*/*z* 191 [QA-H]^−^. The determination of each geometric isomers of the 5-*O*-coumaroylquinic acid was performed considering the chromatographic elution time in reverse-phase chromatography. In particular, 5-*O*-coumaroylquinic acid *trans* isomer eluted before its *cis*-isomer. Therefore, compounds 19 and 21 were tentatively identified as *trans*-5-*O*-coumaroylquinic acid and *cis*-5-*O*-coumaroylquinic acid, respectively [34,39]. Compound 14 showed a [M-H]^−^ ion at *m*/*z* 179 and was identified as caffeic acid. According to the literature data, the base peak ion at *m*/*z* 135 [M-H-CO_2_]^−^ and the fragment ions at *m*/*z* 161 [M-H-H_2_O]^−^ and *m*/*z* 151 [M-H-CO]^−^ highlighted the presence of a carboxylic acid and a phenolic group [35]. Two feruloylquinic acid isomers (compounds 15 and 27) were putatively identified. They displayed a [M-H]^−^ ion at *m*/*z* 367 and characteristic product ions at *m*/*z* 193 and *m*/*z* 191, which represented the fragment ions of ferulic acid and quinic acid, respectively. Compound 15 exhibited a base peak ion at *m*/*z* 193 and was tentatively annotated as 3-*O*-feruloylquinic acid. Instead, compound 27 showed a base peak ion at *m*/*z* 191 and was putatively identified as 5-*O*-feruloylquinic acid [36]. Compound 16 displayed a [M-H]^−^ ion at *m*/*z* 167. The base peak ion at *m*/*z* 123 [M-H-CO_2_]^−^ and the product ions of its tandem mass spectrum at *m*/*z* 152 [M-H-CH_3_]^−^, *m*/*z* 133 [M-H-CH_3_-H_2_O]^−^ and *m*/*z* 108 [M-H-CH_3_-CO_2_]^−^ suggested the presence of a carboxylic acid and a methyl group. According to the mass fragmentation pattern, compound 16 was identified as vanillic acid [37]. Three methyl-caffeoylquinate isomers (compounds 17, 22 and 31) were putatively detected. They showed a [M-H]^−^ ion at *m*/*z* 367 and two characteristic fragment ions at *m*/*z* 179 [M-H-QA-CH_3_]^−^ and *m*/*z* 161 [M-H-QA-CH_3_-H_2_O]^−^. Other representative fragments include the ions at *m*/*z* 191 [QA-H]^−^ and at *m*/*z* 135 [M-H-QA-CH_3_-CO_2_]^−^. This fragmentation pattern confirmed the presence of the quinic acid and the methyl ester scaffolds. Therefore, methyl-3-*O*-caffeoylquinate (17) and methyl-4-*O*-caffeoylquinate (22) can be annotated by the base peak ion at *m*/*z* 161 while methyl-5-*O*-caffeoylquinate (31) is characterized by an intense fragment ion at *m*/*z* 179 [38]. Compound 20 showed a [M-H]^−^ ion at *m*/*z* 197. The base peak ion at *m*/*z* 153 [M-H-CO_2_]^−^ and the fragment ions of their tandem mass spectra at *m*/*z* 179 [M-H-H_2_O]^−^, *m*/*z* 169 [M-H-CO]^−^ and *m*/*z* 161 [M-H-2H_2_O] suggested the presence of a carboxylic acid and a phenolic group. In agreement with the literature data, compound 20 was identified as syringic acid [40]. Three caffeic acid derivatives (compounds 23, 25 and 26) were detected. They presented a [M-H]^−^ ion at *m*/*z* 335 and a base peak ion at *m*/*z* 161 [M-H-SA-H_2_O]^−^, due to the loss of water and shikimic acid moiety. The fragment ions at *m*/*z* 179 [M-H-SA]^−^ and *m*/*z* 135 [M-H-SA-CO_2_]^−^ were detected in the fragmentation pattern of caffeic acid (14). Therefore, these compounds were tentatively identified as three caffeoylshikimic acid isomers [41]. Compound 29 showed a [M-H]^−^ ion at *m*/*z* 163 and a base peak ion at *m*/*z* 119 [M-H-CO_2_]^−^. The fragment ions at *m*/*z* 145 [M-H-H_2_O]^−^, *m*/*z* 135 [M-H-CO]^−^ and *m*/*z* 93 [M-H-CO_2_-C_2_H_2_]^−^ suggested the presence of a phenolic group and the hydroxycinnamic acid moiety. According to the mass fragmentation pattern, compound 29 was annotated as p-coumaric acid [35]. Compound 34 displayed a [M-H]^−^ ion at *m*/*z* 193 and a base peak ion at *m*/*z* 149 [M-H-CO_2_]^−^. The base peak ion and the fragment ions at *m*/*z* 178 [M-H-CH_3_]^−^, *m*/*z* 160 [M-H-CH_3_-H_2_O]^−^ and *m*/*z* 134 [M-H-CH_3_-CO_2_]^−^ indicated the presence of a free carboxylic acid and a methyl group. Based on the literature data, compound 34 was identified as ferulic acid [35]. Two dicaffeoylquinic acid isomers (compounds 42 and 46) were putatively identified and gave a [M-H]^−^ ion at *m*/*z* 515. The base peak ion at *m*/*z* 353 [M-H-CA]^−^, due to the loss of caffeic acid unit, and the fragment ions at *m*/*z* 335 [M-H-CA-H_2_O]^−^ and *m*/*z* 179 [M-H-CA-QA]^−^ indicated the presence of two units of caffeic acid and one unit of quinic acid. Furthermore, the fragmentations of the two compounds followed the fragmentation pattern of chlorogenic acid and neochlorogenic acid (compounds 6 and 13) and was in agreement with the literature data [48].

#### 3.1.2. Flavans Identification

Flavans are a class of 2-phenylchroman compounds largely distributed in plants. The main flavan class is represented by the flavan-3-ols, which include monomeric derivatives, such as (+)-catechin and (−)-epicatechin and oligomeric and polymeric compounds named procyanidins. Procyanidins can be divided into A-type or B-type based on the linkage between monomers. While B-type linkage procyanidins dimers show an interflavan single carbon–carbon bond between monomers, A-type linkage ones are characterized by two linkages between flavan units, an interflavan single carbon–carbon bond and an ether bond [51]. The ether linkage connects the monomers by a six-membered ring, resulting in a difference of 2 Da units. Procyanidins give a characteristic fragmentation pattern, which includes three types of fragmentation mechanisms: the heterocyclic ring fission (HRF), the retro Diels–Alder fission (RDA) and the quinone methide cleavage (QM). The heterocyclic ring fission (HRF) is given by the loss of a phloroglucinol unit (−126 Da), preserving the interflavanic bond between the two monomers. The quinone methide cleavage (QM) represents the fragmentation of the interflavanic bond between two monomers. Therefore, the resulting ions indicate the number of monomers in the oligomeric compounds [31]. Retro Diels–Alder fission (RDA) is given by elimination of hydroxyvinyl benzenediol unit (−152 Da). Three procyanidins dimer B-type linkage (compounds 7, 8 and 24) were detected. They showed a [M-H]^−^ ion at *m*/*z* 577 and a base peak ion at *m*/*z* 425 [M-H-C_8_H_8_O_3_]^−^ due to the RDA fission. HRF fragmentation gave a fragment ion at *m*/*z* 451 [M-H-C_6_H_6_O_3_]^−^ while the QM cleavage produced two fragments at *m*/*z* 289 [M-H-C_15_H_12_O_6_]^−^ and *m*/*z* 287 [M-H-C_15_H_14_O_6_]^−^ [30,31]. Catechin (9) and epicatechin (28) were identified and gave a [M-H]^−^ ion at *m*/*z* 289. The base peak ion at *m*/*z* 245 [M-H-_2_H_4_O]^−^ and the fragment ions at *m*/*z* 271 [M-H-H_2_O]^−^, *m*/*z* 205 [M-H-C_4_H_4_O_2_]^−^ and *m*/*z* 137 [M-H-C_8_H_8_O_3_]^−^, due to the RDA fission, agreed with the literature data [30,32]. Four procyanidins trimer B-type linkage (compounds 12, 30, 32 and 33) were putatively detected. They displayed a [M-H]^−^ ion at *m*/*z* 865 and a base peak ion at *m*/*z* 695 [M-H-C_8_H_8_O_3_-H_2_O]^−^, which is the product of RDA fission and the loss of a molecule of water. Other fragment ions were at *m*/*z* 739 [M-H-C_6_H_6_O_3_]^−^, *m*/*z* 713 [M-H-C_8_H_8_O_3_]^−^ and *m*/*z* 287 [M-H-C_30_H_26_O_12_]^−^, which derived from the HRF fragmentation, the RDA fission and the QM cleavage, respectively [31]. One procyanidin dimer A-type linkage (44) was tentatively detected and gave a [M-H]^−^ ion at *m*/*z* 575. The base peak ion at *m*/*z* 449 [M-H-C_6_H_6_O_3_]^−^, derived from the HRF cleavage, and the product ions at *m*/*z* 557 [M-H-H_2_O]^−^, *m*/*z* 431 [M-H-C_6_H_6_O_3_-H_2_O]^−^ and *m*/*z* 285 [M-H-C_15_H_14_O_6_]^−^, due to the QM cleavage, are in accordance with the fragmentation reported in the literature [30,31]. Flavanones show a different fragmentation pattern than flavan-3-ols. In fact, these compounds occur in the plants as aglycone or glycosides and exhibit a characteristic fragmentation pattern, with prominent fragment ions due to the loss of sugars and retro Diels–Alder fission (RDA) [45]. Eriodyctiol *O*-hexoside (35) showed a [M-H]^−^ ion at *m*/*z* 449 and a base peak ion at *m*/*z* 287 [M-H-Hex]^−^ due to the loss of hexoside unit. Secondary fragments included the *m*/*z* 431 [M-H-H_2_O]^−^, *m*/*z* 151 [M-H-Hex-C_8_H_8_O_2_]^−^, which derived from the RDA fragmentation, and *m*/*z* 135 [M-H-Hex-C_7_H_4_O_4_] [42]. Compound 37 exhibited a [M-H]- ion at *m*/*z* 271. The base peak ion at *m*/*z* 151 [M-H-C_8_H_8_O]^−^, due to the RDA pattern, and the fragment ions at *m*/*z* 253 [M-H-H_2_O]^−^, *m*/*z* 227 [M-H-CO_2_]^−^ and *m*/*z* 107 [M-H-C_8_H_8_O-CO_2_]^−^ allowed to identify the flavanone scaffold. Based on the literature data, compound 37 was identified as naringenin [45]. Compound 41 showed a [M-H]^−^ ion at *m*/*z* 433, a base peak ion at *m*/*z* 271 [M-H-Hex]^−^ and was putatively identified as naringenin *O*-hexoside. The compound exhibited other fragment ions at *m*/*z* 415 [M-H-H_2_O]^−^, *m*/*z* 313 [M-H-C_4_H_8_O_4_]^−^ and *m*/*z* 253 [M-H-Hex-H_2_O]^−^ [47].

#### 3.1.3. Flavonols Identification

Flavanols are a class of 3-hydroxy-2-phenylchromen-4-one polyphenols and exhibit a flavanone-like fragmentation pattern. The aglycon ion represents the main fragment peak due to the cleavage of sugars, but other fragments include the neutral losses of H_2_O, CO and CO_2_ and retro Diels–Alder fission (RDA). A typical fragment ion is represented by the *m*/*z* 179 due to the RDA fragmentation [44]. However, the glycosylated flavonols displayed two prominent fragment ions due to the loss of glucidic units and the cross-ring cleavage of the sugar moiety with a neutral loss of C_4_H_8_O_4_ (120 Da). Rutin (36) showed a [M-H]^−^ ion at *m*/*z* 609 and a base peak ion at *m*/*z* 301 [M-H-Glu-Rha]^−^ that derived from the loss of disaccharide unit. Other fragments are the *m*/*z* 591 [M-H-H_2_O]^−^, *m*/*z* 463 [M-H-Rha]^−^ and *m*/*z* 179 [M-H-Glu-Rha-C_7_H_6_O_2_]^−^, due to the RDA fragmentation. The fragments at 463 (−146 Da) and 301 (−308 Da) allowed identifying the sugars of the disaccharide moiety as glucose and rhamnose [43,44]. Two quercetin *O*-hexoside isomers (compounds 38 and 39) were putatively detected. They displayed a [M-H]^−^ ion at *m*/*z* 463 and a base peak ion at *m*/*z* 301 [M-H-Hex]^−^ due to the loss of hexoside moiety. The main fragments exhibit molecular ions at *m*/*z* 445 [M-H-H_2_O]^−^, *m*/*z* 343 [M-H-C_4_H_8_O_4_]^−^ and *m*/*z* 179 [M-H-Hex-C_7_H_6_O_2_]^−^ that derived from the RDA fragmentation [44]. Three kaempferol derivative isomers (40, 47 and 48) were tentatively identified and gave a [M-H]^−^ ion at *m*/*z* 593. The base peak ion at *m*/*z* 285 [M-H-Hex-Pent]^−^ and the fragment ions at *m*/*z* 447 [M-H-Pent]^−^, *m*/*z* 327 [M-H-Pent-C_4_H_8_O_4_]^−^ and *m*/*z* 257 [M-H-Hex-Pent-CO]^−^ indicate the presence of the disaccharide rutinose and the aglycone kaempferol. Based on the literature data, these compounds were identified as kaempferol *O*-rutinoside isomers [46]. Kaempferol 3-*O*-glucoside (43) showed a [M-H]^−^ ion at *m*/*z* 447 and a base peak ion at *m*/*z* 285 [M-H-Glu]^−^ for the loss of glucoside unit. The fragment ions at *m*/*z* 429 [M-H]^−^, *m*/*z* 327 [M-H-C_4_H_8_O_4_]^−^ and *m*/*z* 255 [M-H-Glu-CH_2_O]^−^ confirmed the identity of the compound and are in accordance with the literature data [46]. Compound 45 displayed a [M-H]^−^ ion at *m*/*z* 301 and a prominent fragment ion at *m*/*z* 179 [M-H-C_7_H_6_O_2_]^−^, due to the RDA fragmentation. Other fragment ions of the tandem mass spectrum included the *m*/*z* 273 [M-H-CO]^−^, *m*/*z* 257 [M-H-CO_2_]^−^ and *m*/*z* 151 [M-H-C_8_H_6_O_3_]^−^. According to the mass fragmentation pattern, compound 45 was identified as quercetin [49].

### 3.2. Quantitative Polyphenols Analysis by HPLC-DAD-FLD

Chromatographic analysis for the quantification of TN polyphenolic composition was conducted as reported in Section 2.3. The HPLC-DAD-FLD analysis allowed the quantification of 19 different compounds, including flavanols, procyanidin compounds, phenolic acids and flavonol derivatives. The results are reported in Table 2.

### 3.3. Total Polyphenols and In Vitro Antioxidant Activity of Thinned Nectarine

In order to obtain an overview of the total polyphenolic content, Folin–Ciocalteau’s assay was performed on hydroalcoholic TN extract. The TN sample exhibited a total phenol content of 17.01 ± 0.35 mg GAE/g of extract. Additionally, the antioxidant activity was evaluated by using DPPH, ABTS and FRAP assays. As reported in Table 3, results were expressed as µmol of TE per g of dried extract.

In order to standardize the results from various studies, DPPH and ABTS assays were also reported as EC_50_, which is the amount of antioxidant necessary to decrease the concentration of the initial solution by 50% [18]. As displayed in Figure 1, TN extract exhibited an EC_50_ of 1.57 mg/mL for DPPH assay and 1.58 mg/mL for ABTS assay.

### 3.4. In Vitro Antidiabetic Activity

#### 3.4.1. α-Amylase Inhibitory Assay

The α-amylase inhibition activity of TN was tested by an in vitro assay. Figure 2 shows the percentage inhibition as IC_50_, which is the amount of compound necessary to inhibit the enzyme activity by 50% [23]. The results revealed that both acarbose and the TN inhibited α-amylase activity in a concentration-dependent manner. The IC_50_ values generated from the percentage inhibition revealed a result of 8.11 and 0.43 mg/mL for TN and acarbose, respectively.

#### 3.4.2. Advanced Glycation End-Product (AGE) Inhibition Assay

The inhibitory activity of TN on AGEs formation was also investigated. The assay is based on the inhibition of specific fluorescence generated during the course of glycation and AGEs formation. Figure 3 shows the percentage inhibition as EC_50_, which is the concentration required to obtain a 50% effect. Under our experimental conditions, TN and rutin produced a concentration-dependent inhibition of AGE, with an EC_50_ of 11.0 and 0.1 mg/mL for TN and rutin, respectively (Figure 3).

## 4. Discussion

Polyphenols, a family of plant secondary metabolites naturally occurring in fruits, are considered critical not only for fruit quality but also for human health benefits [52]. Scientific data showed that polyphenolic compounds contained in nectarine and peach could prevent cellular oxidative stress resulting from free radicals [53]. Interestingly, it was shown that thinned young fruits might exhibit significantly higher antioxidant capacity than those of their ripe counterparts. This was mainly due to the 5–10 times higher total polyphenol content found in thinned young fruits compared to ripe fruits [54]. In the present work, HPLC analyses allowed us to tentatively identify 48 polyphenolic compounds, 19 of which were quantified with analytical standards, confirming the high polyphenolic content of these waste-by products. The quantitative analysis of the polyphenolic profile of TN extract was in line with the available literature data. Specifically, Guo et al. displayed that chlorogenic acid and neochlorogenic acid represent the main polyphenols of unripe nectarines, with a content in the matrix of 100–500 µg/g and 270–1250 µg/g, respectively [54].

Data obtained in this work demonstrated a concentration-dependent inhibitory activity of TN extract on the α-amylase enzyme, with results expressed as IC_50_. It is well-known that the IC_50_ of an inhibitor is very dependent on the assay conditions, such as enzyme concentration and origin, substrate type and concentration, reaction duration, temperature and pH [55]. This makes data comparison with the literature a difficult task. However, by utilizing acarbose as a benchmark, a comparison of the general inhibitory trend could be achieved. Recently, long-term excessive intake of starchy food has been reported to be one of the reasons for hyperglycemia that can even lead to type II diabetes disease [56]. In this regard, reducing the hydrolysis rate of starch through inhibiting digestive enzymes is one suggested way of relieving postprandial hyperglycemia [57]. Of note, the same test conducted with a standard of abscisic acid did not show any inhibitory capacity on the enzyme (data not shown), highlighting that the inhibitory activity of our sample would not be ascribable to the ABA content but to the contribution of other bioactive compounds. In this regard, the role of polyphenols in modulating starch digestion and glycemic levels was widely investigated [57]. As reported by the scientific literature, compounds such as quinic acid derivatives and mono and diglycosyl flavonols (e.g., neochlorogenic acid, chlorogenic acid and rutin) are closely correlated to the inhibition of α-amylase [58]. Therefore, the high content of these polyphenols may justify TN inhibitory efficacy on this enzyme.

Moreover, concentration-dependent inhibition of AGEs formation was observed after testing TN extract through an opportune in vitro assay. AGEs are proteins or fats combined with blood sugars after exposure to a glycation process through the Maillard reaction [59]. These compounds result in being highly stable and resistant to enzymatic degradation, leading to their high accumulation in different tissues, modification in cells and tissues, progressive deterioration of structural integrity and physiological function across multiple organs and increased risk of death [60]. Regarding diabetes mellitus-related hyperglycemic conditions, it is well known that excess intracellular glucose is converted to sorbitol by the polyol pathway, mainly in tissues and organs with an insulin-independent glucose uptake (e.g., retina, peripheral nerves, kidney, erythrocytes). This signaling pathway often results in a complex cascade of events that can culminate in tissue and vascular damage, significantly contributing to the onset of diabetes chronic complications [61]. Therefore, the inhibition of AGEs formation would represent a useful tool for the prevention of diabetes complications. In this regard, growing evidence highlighted that polyphenols are able to prevent AGEs production [62], reasonably supporting the herein observed beneficial activity exerted by TN-based formulation. Polyphenols’ antiglycation properties are mainly due to the inhibition of early Maillard reaction products, especially reactive dicarbonyl as methylglyoxal (MGO) [63]. Phenolic acids and flavans (e.g., gallic acid, p-coumaric acid and epicatechin) can directly reduce the carbonyl groups by a redox reaction, inhibiting the formation of advanced Maillard products. Differently, flavonols (e.g., quercetin, rutin) react with the MGO dicarbonyl moiety, indirectly preventing the formation of glycation products [63,64]. Therefore, it is possible to hypothesize that TN extract may exert an antiglycation action with different mechanism of actions, due to the complexity of polyphenols fraction.

Considering the well-known link between diabetes and oxidative stress and considering the vegetal nature of our food matrix is highly likely to contain antioxidant polyphenols, the attention was focused on the investigation of in vitro antioxidant potential of TN. For these reasons, Folin–Ciocâlteu, DPPH, ABTS and FRAP assays were carried out on a polyphenolic extract of TN. The obtained results are in line with available studies conducted on immature nectarine, confirming the high antioxidant potential of these by-product matrices [25]. In this regard, a study conducted by Guo et al. reported that the antioxidant capacity of peaches and nectarines evaluated by DPPH, ABTS and FRAP assays were 1.3–11.2-fold higher in thinned young fruit compared to ripe fruit [54]. Notable, increasing evidence from in vitro and clinical studies suggests that oxidative stress plays a pivotal role in the pathogenesis of both types of diabetes mellitus. Abnormally high levels of free radicals and the simultaneous decline of antioxidant defense mechanisms can indeed lead to damage to biological structures. These consequences of oxidative stress can promote the development of complications of diabetes mellitus [12]. Similarly, persistent hyperglycemia is recognized as one of the main causes of oxidative stress, supporting a direct cause and effect relationship between hyperglycemia and oxidative stress [13]. In the light of these considerations, the concomitant presence of bioactive molecules with antidiabetic and antioxidant potential in TN-based formulation would further support its supplementation for the management of diabetic pathology. Moreover, thinned unripe fruits revaluation fits properly with the concept of green economy and environmental sustainability, opening new paths for food by-products revaluation [11].

## 5. Conclusions

In conclusion, our data would reasonably support TN as an innovative and promising nutraceutical formulation with beneficial health effects, especially regarding the management of glucose homeostasis. According to previous data obtained from our research group, these beneficial actions may be related to the role of ABA occurred in TN, particularly relating to the insulin-sparing mechanism of action. Furthermore, the concomitant presence of other bioactive components in TN-based nutraceutical formulation, such as polyphenols, may contribute to the management of diabetes-related oxidative stress conditions. Overall, the herein obtained results could represent a starting point for further in-depth animal-based studies and clinical trials aimed at evaluating the antidiabetic effects of the TN-based nutraceutical formulation.

## Figures and Tables

**Figure 1 foods-11-01010-f001:**
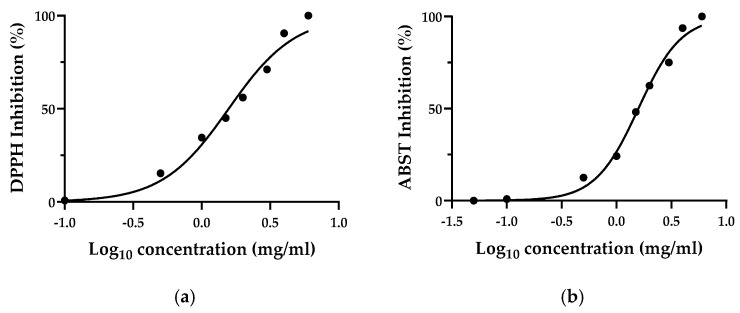
Antioxidant activity of Thinned Nectarine extracts expressed as (**a**) EC_50_ of DPPH assay and (**b**) EC_50_ of ABTS assay. Values represent mean ± standard deviation of triplicate readings.

**Figure 2 foods-11-01010-f002:**
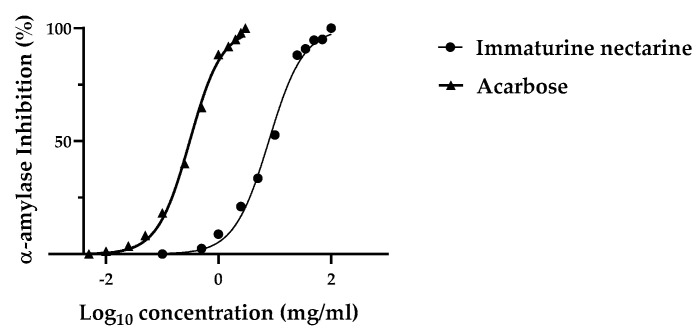
Inhibition of a-amylase activity (%) by Thinned Nectarine and acarbose. Values represent mean ± standard deviation of triplicate readings.

**Figure 3 foods-11-01010-f003:**
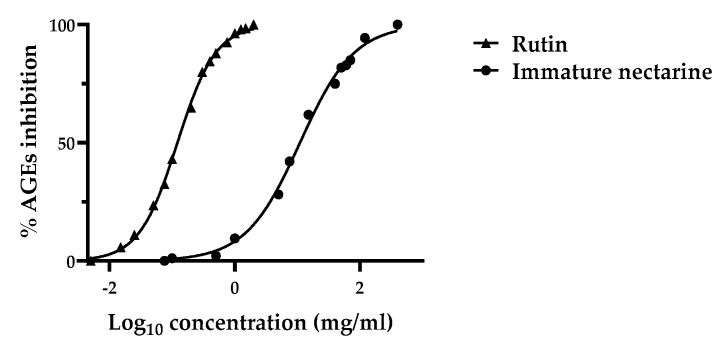
Inhibition of Advances Glycation End-Product formation (%) by Thinned Nectarine and rutin. Values represent mean ± standard deviation of triplicate readings.

**Table 1 foods-11-01010-t001:** Polyphenolic composition of thinned nectarines extracts determined by HPLC-HESI-MS/MS analysis.

Number	Compound	*m*/*z*	Diagnostic Fragmentation	Reference
1	Hydroxycaffeic acid	195.01	177.12 [M-H-H_2_O]^−^–167.08 [M-H-CO]^−^– 151.00 [M-H-CO_2_]^−^–133.10 [M-H-CO_2_-H_2_O]^−^	[25]
2	Dihydroxybenzoic acid	152.97	134.93 [M-H-H_2_O]^−^–124.97 [M-H-CO]^−^– 108.95 [M-H-CO_2_]^−^–96.99 [M-H-2CO]^−^	[25]
3	Gallic acid	169.07	151.20 [M-H-H_2_O]^−^–140.93 [M-H-CO]^−^– 124.85 [M-H-CO_2_]^−^	[26]
4	Homovanillic acid *O*-hexoside	343.01	325.13 [M-H-H_2_O]^−^–298.99 [M-H-CO_2_]^−^– 297.06 [M-H-CO-H_2_O]^−^–181.21 [M-H-Hex]^−^	[27]
5	Quinic acid	190.97	172.95 [M-H-H_2_O]^−^–154.94 [M-H-2H_2_O]^−^– 146.82 [M-H-CO_2_]^−^–110.83 [M-H-CO_2_-2H_2_O]^−^	[28]
6	Neochlorogenic acid	353.30	334.89 [M-H-H_2_O]^−^–191.02 [QA-H]^−^– 178.98 [CA-H]^−^–134.85 [CA-H-CO_2_]^−^	[29]
7	Procyanidin B1	577.14	451.23 [M-H-C_6_H_6_O_3_]^−^–425.17 [M-H-C_8_H_8_O_3_]^−^– 288.98 [M-H-C_15_H_12_O_6_]^−^–287.07 [M-H-C_15_H_14_O_6_]^−^	[30,31]
8	Procyanidin B3	577.45	451.19 [M-H-C_6_H_6_O_3_]^−^–425.25 [M-H-C_8_H_8_O_3_]^−^– 289.04 [M-H-C_15_H_12_O_6_]^−^–287.09 [M-H-C_15_H_14_O_6_]^−^	[30,31]
9	Catechin	289.12	271.03 [M-H-H_2_O]^−^–245.02 [M-H-C_2_H_4_O]^−^– 205.00 [M-H-C_4_H_4_O_2_]^−^–137.00 [M-H-C_8_H_8_O_3_]^−^	[30,32]
10	Coumaric acid *O*-hexoside	325.15	307.26 [M-H-H_2_O]^−^–289.03 [M-H-2H_2_O]^−^– 162.98 [M-H-Hex]^−^–118.97 [M-H-Hex-CO_2_]^−^	[33]
11	3-*O*-Coumaroylquinic acid	337.25	301.77 [M-H-2H_2_O]^−^–191.02 [QA-H]^−^– 173.02 [QA-H-H_2_O]^−^–162.92 [M-H-QA]^−^	[34]
12	Procyanidin C-type	865.25	739.34 [M-H-C_6_H_6_O_3_]^−^–713.20 [M-H-C_8_H_8_O_3_]^−^– 695.19 [M-H-C_8_H_8_O_3_-H_2_O]^−^–286.94 [M-H-C_30_H_26_O_12_]^−^	[31]
13	Chlorogenic acid	353.15	335.24 [M-H-H_2_O]^−^–190.97 [QA-H]^−^– 179.07 [CA-H]^−^–134.98 [CA-H-CO_2_]^−^	[29]
14	Caffeic acid	178.92	160.96 [M-H-H_2_O]^−^–150.80 [M-H-CO]^−^– 134.95 [M-H-CO_2_]^−^–106.86 [M-H-CO-CO_2_]^−^	[35]
15	3-*O*-Feruloylquinic acid	367.18	193.04 [M-H-QA]^−^–191.13 [QA-H]^−^– 173.02 [QA-H-H_2_O]^−^–134.02 [M-H-QA-CH_3_-CO_2_]^−^	[36]
16	Vanillic acid	167.14	151.90 [M-H-CH_3_]^−^–132.92 [M-H-CH_3_-H_2_O]^−^– 122.96 [M-H-CO_2_]^−^–107.90 [M-H-CH_3_-CO_2_]^−^	[37]
17	Methyl 3-*O*-Caffeoylquinate	367.25	335.09 [M-H-2H_2_O]^−^–191.10 [QA-H]^−^– 161.00 [M-H-QA-CH_3_-H_2_O]^−^–134.97 [M-H-QA-CH_3_-CO_2_]^−^	[38]
18	4-*O*-Coumaroylquinic acid	337.12	191.13 [QA-H]^−^–172.86 [QA-H-H_2_O]^−^– 162.98 [M-H-QA]^−^–111.03 [QA-H-2H_2_O]^−^	[34]
19	*Trans*-5-*O*-Coumaroylquinic acid	337.20	190.96 [QA-H]^−^–173.02 [QA-H-H_2_O]^−^– 162.94 [M-H-QA]^−^–111.05 [QA-H-2H_2_O]^−^	[34,39]
20	Syringic acid	196.99	179.05 [M-H-H_2_O]^−^–168.97 [M-H-CO]^−^– 160.82 [M-H-2H_2_O]^−^–152.99 [M-H-CO_2_]^−^	[40]
21	*Cis*-5-*O*-Coumaroylquinic acid	337.13	190.98 [QA-H]^−^–172.84 [QA-H-H_2_O]^−^– 163.05 [M-H-QA]^−^–145.02 [M-H-QA-H_2_O]^−^	[34,39]
22	Methyl 4-*O*-Caffeoylquinate	367.18	296.85 [M-H-C_3_H_2_O_2_]^−^–191.13 [QA-H]^−^– 160.98 [M-H-QA-CH_3_-H_2_O]^−^–134.92 [M-H-QA-CH_3_-CO_2_]^−^	[38]
23	Caffeoylshikimic acid isomer 1	335.18	317.03 [M-H-H_2_O]^−^–179.01 [M-H-SA]^−^– 161.04 [M-H-SA-H_2_O]^−^–134.93 [M-H-SA-CO_2_]^−^	[41]
24	Procyanidin B2	577.25	451.13 [M-H-C_6_H_6_O_3_]^−^–425.07 [M-H-C_8_H_8_O_3_]^−^– 289.10 [M-H-C_15_H_12_O_6_]^−^–287.14 [M-H-C_15_H_14_O_6_]^−^	[30,31]
25	Caffeoylshikimic acid isomer 2	335.12	317.32 [M-H-H_2_O]^−^–178.99 [M-H-SA]^−^– 160.90 [M-H-SA-H_2_O]^−^–134.98 [M-H-SA-CO_2_]^−^	[41]
26	Caffeoylshikimic acid isomer 3	335.17	317.09 [M-H-H_2_O]^−^–179.04 [M-H-SA]^−^– 161.01 [M-H-SA-H_2_O]^−^–134.88 [M-H-SA-CO_2_]^−^	[41]
27	5-*O*-Feruloylquinic acid	367.19	349.23 [M-H-H_2_O]^−^–190.95 [QA-H]^−^– 172.97 [QA-H-H_2_O]^−^–134.09 [M-H-QA-CH_3_-CO_2_]^−^	[36]
28	Epicatechin	288.92	271.06 [M-H-H_2_O]^−^–245.06 [M-H-C_2_H_4_O]^−^– 205.06 [M-H-C_4_H_4_O_2_]^−^–136.94 [M-H-C_8_H_8_O_3_]^−^	[30,32]
29	*p*-Coumaric acid	162.88	144.92 [M-H-H_2_O]^−^–134.83 [M-H-CO]^−^– 118.97 [M-H-CO_2_]^−^–93.08 [M-H-CO_2_-C_2_H_2_]^−^	[35]
30	Procyanidin C-type linkage	865.45	739.33 [M-H-C_6_H_6_O_3_]^−^–713.18 [M-H-C_8_H_8_O_3_]^−^– 695.28 [M-H-C_8_H_8_O_3_-H_2_O]^−^–287.11 [M-H-C_30_H_26_O_12_]^−^	[31]
31	Methyl 5-*O*-Caffeoylquinate	367.19	191.02 [QA-H]^−^–179.00 [M-H-QA-CH_3_]^−^– 161.02 [M-H-QA-CH_3_-H_2_O]^−^–135.00 [M-H-QA-CH_3_-CO_2_]^−^	[38]
32	Procyanidin C1	865.32	739.19 [M-H-C_6_H_6_O_3_]^−^–713.19 [M-H-C_8_H_8_O_3_]^−^– 695.25 [M-H-C_8_H_8_O_3_-H_2_O]^−^–287.04 [M-H-C_30_H_26_O_12_]^−^	[31]
33	Procyanidin C-type linkage	865.22	739.19 [M-H-C_6_H_6_O_3_]^−^–713.26 [M-H-C_8_H_8_O_3_]^−^– 695.32 [M-H-C_8_H_8_O_3_-H_2_O]^−^–287.11 [M-H-C_30_H_26_O_12_]^−^	[31]
34	Ferulic acid	193.16	177.95 [M-H-CH_3_]^−^–160.02 [M-H-CH_3_-H_2_O]^−^– 148.94 [M-H-CO_2_]^−^–133.94 [M-H-CH_3_-CO_2_]^−^	[35]
35	Eriodyctiol *O*-hexoside	449.20	431.11 [M-H-H_2_O]^−^–287.07 [M-H-Hex]^−^– 150.72 [M-H-Hex-C_8_H_8_O_2_]^−^–135.09 [M-H-Hex-C_7_H_4_O_4_]^−^	[42]
36	Rutin	609.34	591.40 [M-H-H_2_O]^−^–463.22 [M-H-Rha]^−^– 301.16 [M-H-Glu-Rha]^−^–179.06 [M-H-Glu-Rha-C_7_H_6_O_2_]^−^	[43,44]
37	Naringenin	271.04	253.07 [M-H-H_2_O]^−^–226.99 [M-H-CO_2_]^−^– 150.92 [M-H-C_8_H_8_O]^−^–106.92 [M-H-C_8_H_8_O-CO_2_]^−^	[45]
38	Quercetin-3-*O*-glucoside	463.18	445.14 [M-H-H_2_O]^−^–343.04 [M-H-C_4_H_8_O_4_]^−^– 301.03 [M-H-Glu]^−^–179.09 [M-H-Hex-C_7_H_6_O_2_]^−^	[44]
39	Quercetin-*O*-glucoside isomer	463.22	445.17 [M-H-H_2_O]^−^–343.17 [M-H-C_4_H_8_O_4_]^−^– 301.10 [M-H-Hex]^−^–178.97 [M-H-Hex-C_7_H_6_O_2_]^−^	[44]
40	Kaempferol-*O*-rutinoside isomer 1	593.32	447.21 [M-H-Pent]^−^–327.18 [M-H-Pent-C_4_H_8_O_4_]^−^– 285.10 [M-H-Hex-Pent]^−^–257.14 [M-H-Hex-Pent-CO]^−^	[46]
41	Naringenin *O*-hexoside	433.23	415.26 [M-H-H_2_O]^−^–313.23 [M-H-C_4_H_8_O_4_]^−^– 271.11 [M-H-Hex]^−^–253.07 [M-H-Hex-H_2_O]^−^	[47]
42	Dicaffeoylquinic acid isomer 1	515.07	353.17 [M-H-CA]^−^–334.98 [M-H-CA-H_2_O]^−^– 317.22 [M-H-CA-2H_2_O]^−^–178.91 [M-H-CA-QA]^−^	[48]
43	Kaempferol-3-*O*-glucoside	447.14	428.99 [M-H-H_2_O]^−^–327.07 [M-H-C_4_H_8_O_4_]^−^– 285.07 [M-H-Glu]^−^–255.01 [M-H-Glu-CH_2_O]^−^	[46]
44	Procyanidin dimer A-type linkage	575.19	557.19 [M-H-H_2_O]^−^–449.13 [M-H-C_6_H_6_O_3_]^−^– 431.13 [M-H-C_6_H_6_O_3_-H_2_O]^−^–285.11 [M-H-C_15_H_14_O_6_]^−^	[30,31]
45	Quercetin	301.18	273.17 [M-H-CO]^−^–257.07 [M-H-CO_2_]^−^– 179.11 [M-H-C_7_H_6_O_2_]^−^–150.88 [M-H-C_8_H_6_O_3_]^−^	[49]
46	Dicaffeoylquinic acid isomer 2	515.18	353.04 [M-H-CA]^−^–334.98 [M-H-CA-H_2_O]^−^– 317.09 [M-H-CA-2H_2_O]^−^–178.84 [M-H-CA-QA]^−^	[48]
47	Kaempferol-*O*-rutinoside isomer 2	593.34	447.15 [M-H-Pent]^−^–327.12 [M-H-Pent-C_4_H_8_O_4_]^−^– 284.95 [M-H-Hex-Pent]^−^–257.15 [M-H-Hex-Pent-CO]^−^	[46]
48	Kaempferol-*O*-rutinoside isomer 3	593.38	447.13 [M-H-Pent]^−^–327.09 [M-H-Pent-C_4_H_8_O_4_]^−^– 285.11 [M-H-Hex-Pent]^−^–256.83 [M-H-Hex-Pent-CO]^−^	[46]

**Table 2 foods-11-01010-t002:** Quantitative analysis of Thinned Nectarines polyphenols determined by HPLC-DAD-FLD analysis.

Compound	Retention Time (min)	Mean Value ± SD (µg/g)
Gallic acid	4.00	168.31 ± 1.51
Neochlorogenic acid	7.76	1456.98 ± 1.19
Procyanidin B1 + Procyanidin B3 *	12.70	8.41 ± 0.02
Catechin	13.34	128.32 ± 0.36
Chlorogenic acid	13.72	1496.85 ± 0.22
Caffeic acid	13.76	15.85 ± 0.06
Vanillic acid	14.78	19.28 ± 0.91
Syringic acid	17.20	115.16 ± 0.21
Procyanidin B2	18.03	6.55 ± 0.01
Epicatechin	19.54	34.63 ± 0.83
*p*-Coumaric acid	20.65	5.05 ± 0.33
Procyanidin C1	22.47	12.66 ± 0.01
Ferulic acid	24.39	10.59 ± 0.02
Rutin	28.27	48.86 ± 0.67
Naringenin	31.14	10.92 ± 0.42
Quercetin-3-*O*-glucoside	32.68	166.01 ± 3.35
Kaempferol-3-*O*-glucoside	36.84	63.65 ± 3.01
Quercetin	46.06	17.89 ± 0.41

Values are expressed in µg/g ± standard deviation (SD) of three repetitions. * Procyanidins B1 and B3 peaks were partially overlapped and were quantified as a mixture of two compounds using the procyanidin B1 calibration curve.

**Table 3 foods-11-01010-t003:** Antioxidant activity of Thinned Nectarine extracts evaluated by DPPH, ABTS and FRAP assays.

Antioxidant Activity (µmol TE/g TN ± SD)		
**DPPH Assay**	**ABTS Assay**	**FRAP Assay**
40.09 ± 0.14	63.26 ± 0.72	58.07 ± 0.14

The results are expressed as μmol TE per gram of TN extract. Abbreviations: TN, thinned nectarine; DPPH, 2,2-diphenyl-1-picrylhydrazyl; ABTS, 2,2′-azino-bis (3-ethylbenzothiazoline-6-sulfonic acid); FRAP, ferric reducing antioxidant power; TE, Trolox equivalent. Values are mean ± standard deviation (SD) of three repetitions.

## Data Availability

The data used to support the findings of this study are included in the article.

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
