# Peer review of "Thinned Nectarines, an Agro-Food Waste with Antidiabetic Potential: HPLC-HESI-MS/MS Phenolic Characterization and In Vitro Evaluation of Their Beneficial Activities"

_foods, 2022, doi:10.3390/foods11071010_

Round 1

Reviewer 1 Report

Thinned Nectarines, an Agro-Food Waste with antidiabetic potential: HPLC-HESI-MS/MS Phenolic Characterization and In Vitro Evaluation of their Beneficial Activities is very interesting and well written.

Presented manuscript is on good scientific level and represent a very high scientific value manuscript. 

The summary. Authors give a short presentation of manuscript.

Introduction section.

The Introduction section includes all necessary information about examined objects and problems. Formatted aim and main hypotheses are good presented at the end of Introductions' section. The problem described in manuscript is a very new and represent problems of quantity of polyphenols extracted from thinned nectarines and their health potential in case of diabetes.

Materials and method section is well written without any doubts.

Results

Page 12, Table 2. In presented table Authors give results of identified polyphenols.  According to declaration in sub-section about Statistical analysis results are mean with SD, but my doubts are there is no other statistical tools as homogenous groups and p-values for polyphenols data. Please add this information to Table 2.

The numbering of the tables has been duplicated. There are two tables No. 2 in the text, and the table No. 3 is missing - Please change it.

The discussion section presents a good comparison of the obtained results with other results available in the data basis.

Presented conclusions are corresponding with all information presented via Authors’ in manuscript text.

General opinion:  After carefully manuscript reading, I think, that presented manuscript is a very valuable.

Reviewer 2 Report

Dear authors,

Your Manuscript has potential to be interesting for researchers who analyze natural compounds from plant source and promote their applications in pharmacy as replacer for synthetic drugs. Furthermore, this study aimed to promote the valorisation of thinned fruits in general, which fits into the concept of green economy and environmental sustainability.

Your manuscript is well organized and the results are clearly presented and discussed, so I have only one suggestion which can  improve current form of manuscript.

  1. I suggest to the authors that evaluate the correlations between antioxidant/antidiabetic assays and quantified phenolic compounds (determined by HPLC-DAD-FLD analysis), because it can help assess the impact of individual phenolic compounds and confirm certain assumptions that have been interpreted in the "Discussion" section.
